# Dynamic Ultrasound Examination for Extensor Pollicis Longus Tendon Rupture after Palpation-Guided Corticosteroid Injection

**DOI:** 10.3390/diagnostics13050959

**Published:** 2023-03-03

**Authors:** Ying-Chun Chen, Wei-Ting Wu, Kamal Mezian, Vincenzo Ricci, Levent Özçakar, Ke-Vin Chang

**Affiliations:** 1Department of Physical Medicine and Rehabilitation, College of Medicine, National Taiwan University, Taipei 10048, Taiwan; 2Department of Physical Medicine and Rehabilitation, National Taiwan University Hospital, Bei-Hu Branch, Taipei 10845, Taiwan; 3Department of Rehabilitation Medicine, Charles University, First Faculty of Medicine and General University Hospital in Prague, 12800 Prague, Czech Republic; 4Physical and Rehabilitation Medicine Unit, Luigi Sacco University Hospital, ASST Fatebenefratelli-Sacco, 20157 Milan, Italy; 5Department of Physical and Rehabilitation Medicine, Hacettepe University Medical School, Ankara 06100, Turkey; 6Center for Regional Anesthesia and Pain Medicine, Wang-Fang Hospital, Taipei Medical University, Taipei 11600, Taiwan

**Keywords:** extensor tendon, wrist, ultrasonography, blind, intervention

## Abstract

This report aimed to present a case of wrist-tendon rupture and to discuss a rare complication after corticosteroid injection. A 67-year-old woman had difficulty extending her left-thumb interphalangeal joint several weeks after a palpation-guided local corticosteroid injection. Passive motions remained intact without sensory abnormalities. Ultrasound examination showed hyperechoic tissues at the site of the extensor pollicis longus (EPL) tendon at the wrist level and an atrophic EPL muscle stump at the forearm level. Dynamic imaging demonstrated no motion in the EPL muscle during passive thumb flexion/extension. The diagnosis of complete EPL rupture, possibly due to inadvertent intratendinous corticosteroid injection, was therefore confirmed.

This report aimed to present a case of wrist-tendon rupture and to discuss a rare complication after corticosteroid injection. A 67-year-old female complained of difficulty in extending the left thumb for several months. The medical history comprised radial wrist pain after a contusion injury one year ago, for which corticosteroid injection under palpation guidance had also been performed, with gradual symptom relief. The physical examination revealed loss of the active extension of the left-thumb interphalangeal joint with normal passive motions (Figure 1, Appendix A). Other finger joints were not affected, nor were there any sensory abnormalities over the distal upper extremity. Plain radiographs of the left wrist showed no evidence of fracture or joint dislocation. The ultrasound examination of the left wrist showed hyperechoic tissue between the Lister’s tubercle and the extensor digitorum communis tendons where the extensor pollicis longus (EPL) tendon should normally reside (Figure 2A). Herewith, the EPL tendon was clearly visualized at the right wrist (Figure 2B). 

The posterior interosseous nerve (PIN) remained bilaterally intact based on dynamic tracking from the level of the supinator tunnel to the dorsal wrist [1]. Moving the transducer more proximally toward the distal forearm, a hyperechoic region was seen at the ulnar aspect of the first extensor compartment, representing the atrophic EPL muscle (Figure 3A). On the normal side, the intact EPL muscle could be explicitly recognized (Figure 3B). Dynamic ultrasound was applied to trace the left EPL tendon distally to proximally, whereby no excursion was observed for the EPL muscle during passive thumb flexion/extension (Appendix A). As a complete rupture of the left EPL tendon was diagnosed/confirmed, the patient was referred for surgical repair.

Finger drop can ensue due to neurological and orthopedic causes [2]. In cases with more generalized weakness or sensory signs, neurological causes, such as PIN palsy, cervical radiculopathy or multifocal neuropathy, should be considered [3]. More specifically, PIN palsy may result in the loss of finger and thumb extension, without wrist drop. Four types of PIN palsy have been reported. Among them, (although uncommon) type 4 exclusively involves the branch supplying the EPL, with isolated difficulty in extending the IP joint of the thumb [4]. Orthopedic causes, on the other hand, include rupture, subluxation or entrapment of the finger extensor tendon. The absence of the tenodesis effect, i.e., finger automatic extension during wrist passive flexion, would indicate an abnormality of the tendons instead of nerves [2]. In patients with an isolated loss of thumb extension, the EPL tendon (responsible for the extension of the thumb interphalangeal joint) should be investigated. 

Static ultrasound examination can serve as a helpful tool for scrutinizing tendon injury, whereas dynamic imaging can be employed to validate motion problems. The most common sonographic finding of tendon rupture is loss of fiber continuity with the formation of a retracted stump. The absence of normal tendon gliding during passive motion further confirms the diagnosis [5]. In chronic cases, the proximal muscle part would become thinned and hyperechoic due to fat infiltration and disuse atrophy. In the case of a ruptured EPL tendon, the tubular-shaped hypoechoic area might be seen, corresponding to fluid, hemorrhage, and scar tissue [6]. In addition, a bulging contoured hypoechoic lesion (also referred to as a pseudo-mass), located between the Lister’s tubercle and the carpometacarpal joint, has been reported as being more frequent over the ruptured EPL than other finger tendons [5]. 

One of the important etiologies of spontaneous EPL tendon rupture is intratendinous corticosteroid injection. Other causes include distal radial fracture, repetitive friction against metal implants, and contusions [7]. Local corticosteroid injection is associated with rupture, not only in the EPL tendon, but also in other locations, such as the rotator cuff, common extensor, patellar, and Achilles tendons [8,9,10,11,12]. Intratendinous corticosteroid injection in animal models was found to inhibit connective-tissue formation and lead to a reduction of tendon mass and biomechanical integrity [13,14]. Lu et al. [15] reported 13 patients with tendon rupture after corticosteroid injection on the wrist region. The most common injected site was the radial styloid under the clinical impression of De Quervain’s tenosynovitis. The injection times over the same sites ranged from 1 to 4, whereas the periods from the last injection ranged from 3 to 32 weeks.

Needless to say, ultrasound is a preferred tool for guiding the needle, in order to avoid unwanted intratendinous corticosteroid injection [16,17]. Peritendinous or intrabursal corticosteroid injection, even in the highest dose, do not affect the tendon’s elasticity when properly performed under ultrasound guidance [18]. Compared to palpation guidance, ultrasound guidance provides fewer local complications, e.g., fat atrophy and depigmentation [19]. Furthermore, ultrasound- vs. palpation-guided corticosteroid injection has been proven to have better accuracy and clinical outcome in bicipital tendinosis [20]. In short, this case of ours revealed that static/dynamic ultrasound played a vital role in the management of (thumb) tendon disorders. This is true not only for a precise diagnosis or targeting during an intervention, but is therefore also paramount to the indirect avoidance of adverse effects, e.g., an intratendinous injection.

## Figures and Tables

**Figure 1 diagnostics-13-00959-f001:**
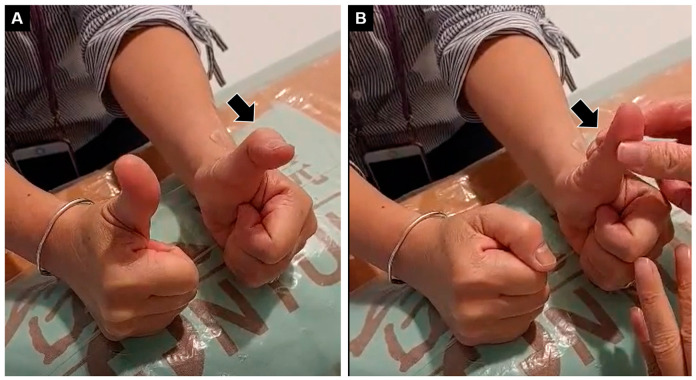
(**A**) Inability to extend the interphalangeal joint of the left thumb (black arrow), with the normal right side. (**B**) Normal passive movement of the same joint.

**Figure 2 diagnostics-13-00959-f002:**
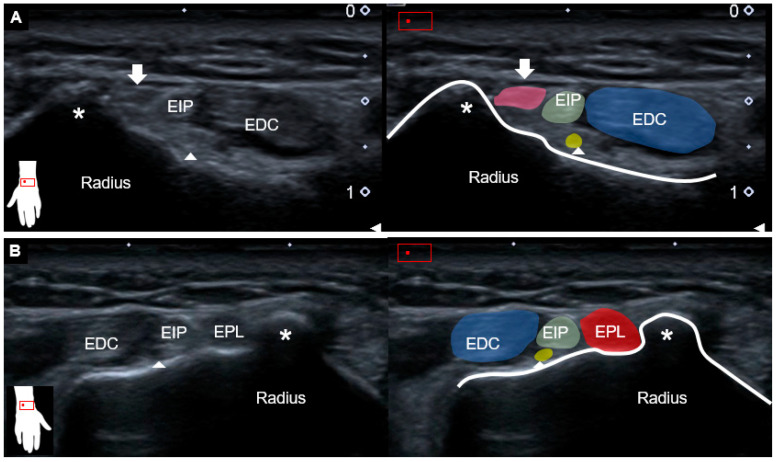
Ultrasound imaging (transverse view) and schematic drawing of (**A**) left and (**B**) right wrists at the level of Lister’s tubercle. Compared to the right side, EPL tendon was absent with hyperechoic fibrofatty tissues (white arrow and pink shade) instead. Asterisks, Lister’s tubercle; EPL and red shade, extensor pollicis longus tendon; EDC and blue shade, extensor digitorum communis tendon; EIP and green shade, extensor indicis proprius tendon; arrowheads and yellow shade, posterior interosseous nerve.

**Figure 3 diagnostics-13-00959-f003:**
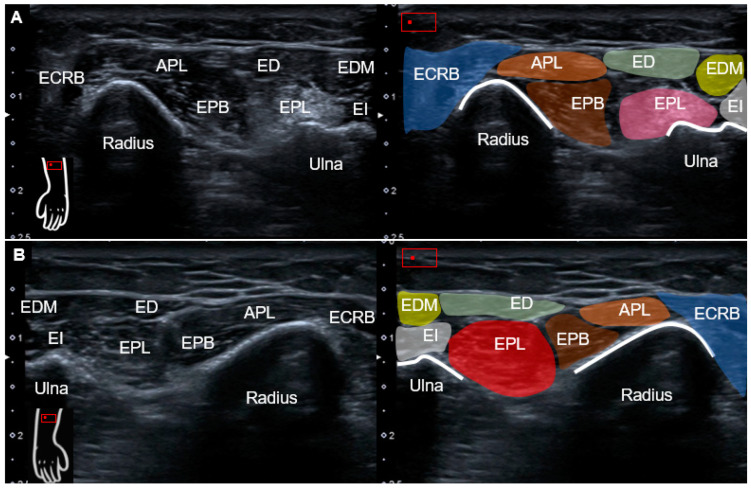
Ultrasound imaging (transverse view) and schematic drawing of the (**A**) left and (**B**) right distal forearm. Compared with the normal side, the left EPL muscle, located at the ulnar aspect of EPB muscle, was hyperechoic and atrophic. APL and orange shade, abductor pollicis longus; ECRB and blue shade, extensor carpi radialis brevis; EPB and brown shade, extensor pollicis brevis; ED and green shade, extensor digitorum communis; EDM and yellow shade, extensor digiti minimi; EI and gray shade, extensor indices; EPL and pink/red shade, extensor pollicis longus.

## Data Availability

Data are contained within the main text of the manuscript.

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
