# Peer review of "Dynamic Ultrasound Examination for Extensor Pollicis Longus Tendon Rupture after Palpation-Guided Corticosteroid Injection"

_diagnostics, 2023, doi:10.3390/diagnostics13050959_

Round 1

Reviewer 1 Report

The authors reported EPL tendon rupture after steroid injection using dynamic ultrasound examination. I think it is a interesting report and has enough quality to be published in Diagnostics.

Author Response

Reviewer 1

Comment:

The authors reported EPL tendon rupture after steroid injection using dynamic ultrasound examination. I think it is an interesting report and has enough quality to be published in Diagnostics.

Response:

We appreciate the kind comment from the reviewer. We are grateful for the reviewer’s effort in reviewing this manuscript.

Reviewer 2 Report

This care report describes a 67 years old woman with rupture of the tendon of the extensor pollicis longus (EPL) muscle. Muscle sonography shows hyperechoic EPL without movement during passive flexion of the thumb.

This case report is almost impossible to understand. Clinical information is only provided in the abstract but not in the text body of the manuscript. It remains unclear for which symptoms or disease steroid injection war performed. It remains further unclear what was the target for the steroid injection.

It is unclear if parts of the text body which explain the abbreviation belong to the figure legend.

The posterior interosseous nerve is only depicted on a single transverse image on the level of the wrist, which is distally to the muscle belly. This image is not sufficient to proof that the nerve is intact. If the palsy was caused by nerve damage, the nerve must have been damaged proximally to the muscle belly. However, this region is not depicted by the video or by the images of the manuscript. If nerve lesions should be excluded, electromyography is required to proof loss of nerve function. 

Author Response

Reviewer 2

Comment:

This care report describes a 67 years old woman with rupture of the tendon of the extensor pollicis longus (EPL) muscle. Muscle sonography shows hyperechoic EPL without movement during passive flexion of the thumb.

This case report is almost impossible to understand. Clinical information is only provided in the abstract but not in the text body of the manuscript. It remains unclear for which symptoms or disease steroid injection war performed. It remains further unclear what was the target for the steroid injection.

It is unclear if parts of the text body which explain the abbreviation belong to the figure legend.

The posterior interosseous nerve is only depicted on a single transverse image on the level of the wrist, which is distally to the muscle belly. This image is not sufficient to proof that the nerve is intact. If the palsy was caused by nerve damage, the nerve must have been damaged proximally to the muscle belly. However, this region is not depicted by the video or by the images of the manuscript. If nerve lesions should be excluded, electromyography is required to proof loss of nerve function. 

Response:

   We appreciate the comment from the reviewer. The editorial office made a mistake during the copyediting process and send the faulty version for review. A whole introductory paragraph and the explanation for the abbreviations in the figures have been missed. The missed paragraph has been added in the revised version as “A 67-year-old female complained of difficulty in extending the left thumb for several months. Medical history comprised radial wrist pain after a contusion injury one year ago, for which corticosteroid injection under palpation guidance had also been performed, with gradual symptom relief. Physical examination revealed loss of active extension of the left thumb interphalangeal joint with normal passive motions (Figure 1, Video 1S). Other finger joints were not affected, nor were there any sensory abnormalities over the distal upper extremity. Plain radiographs of the left wrist showed no evidence of fracture or joint dislocation”.

   The missed abbreviations for the figures are shown as the following:

Figure 2: Asterisks, Lister’s tubercle; EPL and red shade, extensor pollicis longus tendon; EDC and blue shade, extensor digitorum communis tendon; EIP and green shade, extensor indicis proprius tendon; arrowheads and yellow shade, posterior interosseous nerve.

Figure 3: APL and orange shade, abductor pollicis longus; ECRB and blue shade, extensor carpi radialis brevis; EPB and brown shade, extensor pollicis brevis; ED and green shade, extensor digitorum communis; EDM and yellow shade, extensor digiti minimi; EI and gray shade, extensor indices; EPL and pink/red shade, extensor pollicis longus.

   Furthermore, the posterior interosseous nerve was tracked from the supinator tunnel level to the wrist. Therefore, we could confirm the integrity of the posterior interosseous nerve through dynamic tracking. The clarification has been added in the revised manuscript as “The posterior interosseous nerve (PIN) remained intact bilaterally based on dynamic tracking from the level of the supinator tunnel to the dorsal wrist”. The tracking protocol for the PIN has also been referenced: Sonographic tracking of the upper limb peripheral nerves: a pictorial essay and video demonstration. Am J Phys Med Rehabil. 2015 Sep;94(9):740-7.

Reviewer 3 Report

The clinical case presented by the authors can be interesting as a clinical case study. However, the manuscript needs a scientific structure and format to help the reader understand the case better.

The introduction begins directly with two photographs, without any background information on the pathology, which situates the reader on the idea to be discussed, with epidemiological or clinical data to support these arguments. 

There is no justification with current evidence.

There is no chronology of the clinical history, tests, examinations, treatment or intervention. Therefore, the article needs more power and scientific interest.

The manuscript should be completely reworked and provided with the appropriate structure and the correct sections to help the reader to follow the clinical case and its importance.

Author Response

Reviewer 3

Comment:

The clinical case presented by the authors can be interesting as a clinical case study. However, the manuscript needs a scientific structure and format to help the reader understand the case better.

The introduction begins directly with two photographs, without any background information on the pathology, which situates the reader on the idea to be discussed, with epidemiological or clinical data to support these arguments. 

There is no justification with current evidence.

There is no chronology of the clinical history, tests, examinations, treatment or intervention. Therefore, the article needs more power and scientific interest.

The manuscript should be completely reworked and provided with the appropriate structure and the correct sections to help the reader to follow the clinical case and its importance.

Response:

We appreciate the comment from the reviewer. The editorial office made a mistake during the copyediting process and send the faulty version for review. A whole introductory paragraph and the explanation for the abbreviation in the figures have been missed. The missing paragraph has been added in the revised version as “A 67-year-old female complained of difficulty in extending the left thumb for several months. Medical history comprised radial wrist pain after a contusion injury one year ago, for which corticosteroid injection under palpation guidance had also been performed, with gradual symptom relief. Physical examination revealed loss of active extension of the left thumb interphalangeal joint with normal passive motions (Figure 1, Video 1S). Other finger joints were not affected, nor were there any sensory abnormalities over the distal upper extremity. Plain radiographs of the left wrist showed no evidence of fracture or joint dislocation”.

Furthermore, as this article is written in the format of “interesting image”, not a case report. The background/current evidence of this disease is given in the latter part of the manuscript as” Finger drop can ensue due to neurological and orthopedic causes. In cases with more generalized weakness or sensory signs, neurological causes such as PIN palsy, cer-vical radiculopathy or multifocal neuropathy should be considered. More specifically, PIN palsy may result in loss of finger and thumb extension, without wrist drop. Four types of PIN palsy have been reported. Among them, (although uncommon) type 4 exclu-sively involves the branch supplying the EPL, with isolated difficulty in extending the IP joint of the thumb. Orthopedic causes, on the other hand, include rupture, subluxation or entrapment of the finger extensor tendon. Absence of the tenodesis effect, i.e. finger au-tomatic extension during wrist passive flexion, would indicate abnormality of the tendons instead of nerves. In patients with isolated loss of thumb extension, the EPL tendon (re-sponsible for extension of the thumb interphalangeal joint) should be investigated”

 By the way, a paragraph to address the issue/clinical evidence of spontaneous tendon rupture after corticosteroid injection has been given in the revised manuscript as “Lu et al. reported 13 patients with tendon rupture after corticosteroid injection on the wrist region. The most common injected site was the radial styloid under the clinical impression of De Quervain's tenosynovitis. The injection times over the same sites ranged from 1 to 4, whereas the periods from the last injection to identification of tendon rupture ranged from 3 to 32 weeks”. We appreciate the kind understanding of the reviewer.

Reviewer 4 Report

Dear Editor and Authors,

Thank you for the opportunity to review the manuscript entitled “Dynamic Ultrasound Examination for Extensor Pollicis Longus Tendon Rupture After Palpation-guided Corticosteroid Injection”. The authors presented a case of complete EPL rupture, possibly due to inadvertent intratendinous corticosteroid injection. The case is interesting and should be of interest for the Journal readers. However, I have some remarks:

-          The aim should be clearly specified in the abstract and in the text, e.g. “to present a case of…and discuss a rare complication…”|

-          Although the type of manuscript is not a typical case report, but Interesting Images, I suggest starting the article with a short case presentation to give a background for the image described (the history of the patient, reason for steroid injection, etc).

-It should be shortly stated what is the most common indication of steroid injections in hand and what is the frequency of the described complication and when it usually appears /was the timing of the rupture typical?/

Author Response

Reviewer 4

Comment:

Dear Editor and Authors,

Thank you for the opportunity to review the manuscript entitled “Dynamic Ultrasound Examination for Extensor Pollicis Longus Tendon Rupture After Palpation-guided Corticosteroid Injection”. The authors presented a case of complete EPL rupture, possibly due to inadvertent intratendinous corticosteroid injection. The case is interesting and should be of interest for the Journal readers. However, I have some remarks:

-          The aim should be clearly specified in the abstract and in the text, e.g. “to present a case of…and discuss a rare complication…”|

-          Although the type of manuscript is not a typical case report, but Interesting Images, I suggest starting the article with a short case presentation to give a background for the image described (the history of the patient, reason for steroid injection, etc).

-It should be shortly stated what is the most common indication of steroid injections in hand and what is the frequency of the described complication and when it usually appears /was the timing of the rupture typical?/

Response:

First, we appreciate the kind comment from the reviewer. The editorial office made a mistake during the copyediting process and send the faulty version for review. A whole introductory paragraph and the explanation for the abbreviation in the figures were missed. That’s is why the background of the patient was not described in the original version. The missed paragraph has been added in the revised version as “A 67-year-old female complained of difficulty in extending the left thumb for several months. Medical history comprised radial wrist pain after a contusion injury one year ago, for which corticosteroid injection under palpation guidance had also been performed, with gradual symptom relief. Physical examination revealed loss of active extension of the left thumb interphalangeal joint with normal passive motions (Figure 1, Video 1S). Other finger joints were not affected, nor were there any sensory abnormalities over the distal upper extremity. Plain radiographs of the left wrist showed no evidence of fracture or joint dislocation”.

Second, in the abstract and main text, we have added a sentence in accordance to the reviewer’s suggestion as “This report aimed to present a case of wrist tendon rupture and to discuss a rare complication after corticosteroid injection”.

Third, regarding the issue of corticosteroid injection for wrist joint, we would like to reference an original study enrolling 13 patients with tendon rupture after corticosteroid injection on the wrist region (The clinical effect of tendon repair for tendon spontaneous rupture after corticosteroid injection in hands. Medicine (Baltimore). 2016 Oct; 95(41): e5145). The most common injected site was the radial styloid under the clinical impression of De Quervain's tenosynovitis. The injection times over the same sites ranged from 1 to 4, whereas the periods from the last injection to identification of tendon rupture ranged from 3 to 32 weeks. The aforementioned information has been added in the revised manuscript as “Lu et al. reported 13 patients with tendon rupture after corticosteroid injection on the wrist region. The most common injected site was the radial styloid under the clinical impres-sion of De Quervain's tenosynovitis. The injection times over the same sites ranged from 1 to 4, whereas the periods from the last injection to identification of tendon rupture ranged from 3 to 32 weeks”.

Round 2

Reviewer 2 Report

The authors improved their manuscript significantly.

Author Response

We appreciate the kind comments from the reviewer.

Reviewer 3 Report

The authors improved the article's presentation with a contextual introduction and new references.

As a minor comment, I think the colored images could also include the acronyms of the corresponding muscle for better understanding.

Author Response

We appreciate the kind comments from the reviewer. The acronyms have been added on the color sonographic images to better understanding. Please kindly refer to revised Figure 2 and 3.